# Protein Intake, Fatigue and Quality of Life in Stable Outpatient Kidney Transplant Recipients

**DOI:** 10.3390/nu12082451

**Published:** 2020-08-14

**Authors:** Antonio W. Gomes Neto, Karin Boslooper-Meulenbelt, Marit Geelink, Iris M. Y. van Vliet, Adrian Post, Monica L. Joustra, Hans Knoop, Stefan P. Berger, Gerjan J. Navis, Stephan J. L. Bakker

**Affiliations:** 1Department of Internal Medicine, University Medical Center Groningen, University of Groningen, 9700 RB Groningen, The Netherlands; a.w.gomes.neto@umcg.nl (A.W.G.N.); m.geelink@umcg.nl (M.G.); a.post01@umcg.nl (A.P.); s.p.berger@umcg.nl (S.P.B.); g.j.navis@umcg.nl (G.J.N.); s.j.l.bakker@umcg.nl (S.J.L.B.); 2Department of Dietetics, University Medical Center Groningen, 9700 RB Groningen, The Netherlands; i.m.y.van.vliet@umcg.nl; 3Interdisciplinary Center for Psychopathology and Emotion Regulation, University Medical Center Groningen, University of Groningen, 9700RB Groningen, The Netherlands; Mo.joustra@zgt.nl; 4Department of Medical Psychology, Amsterdam University Medical Centers, University of Amsterdam, 1000 DD Amsterdam, The Netherlands; hans.knoop@amsterdamumc.nl

**Keywords:** kidney transplantation, fatigue, quality of life, protein, nutrition

## Abstract

Fatigue is a frequent complaint in kidney transplant recipients (KTR), often accompanied by poor quality of life (QoL). The role of nutrition as determinant of fatigue in KTR is largely unexplored. The aims of this study are to examine the association of protein intake with fatigue and QoL in KTR and to identify other determinants of fatigue. This cross-sectional study is part of the TransplantLines Cohort and Biobank Study (NCT03272841). Protein intake was calculated from urinary urea nitrogen (UUN) in 24-h urine samples. Fatigue was assessed by the Checklist Individual Strength (CIS) questionnaire; moderate and severe fatigue were defined as a CIS score of 20–34 and ≥ 35, respectively. QoL was assessed with the RAND-36-Item Health Survey (RAND-36). Associations of protein intake with fatigue and QoL were analyzed using multinomial logistic and linear regression analyses. We included 730 stable outpatient KTR (median age 58 year [IQR 48–65], 57% male) with a mean protein intake of 82.2 ± 21.3 g/d. Moderate and severe fatigue were present in 254 (35%) and 245 (34%) of KTR. Higher protein intake was significantly associated with lower risk of moderate fatigue (OR 0.89 per 10 g/d; 95%CI 0.83–0.98, *p* = 0.01), severe fatigue (OR 0.85; 95%CI 0.78–0.92, *p <* 0.001) and was associated with higher physical component summary score of QoL (β 0.74 per 10 g/d; 95%CI 0.39–1.09, *p* < 0.001). Higher BMI, a history of dialysis, glomerulonephritis as primary kidney disease and a history of combined organ transplantation were also associated with severe fatigue. In conclusion, amongst the potential modifiable factors of fatigue, higher protein intake is independently associated with lower risk of moderate and severe fatigue and with better QoL in KTR. These findings underline the need to incorporate nutritional assessment in the diagnostic work-up of fatigue. Intervention studies are needed to assess the benefits and safety of higher protein intake in KTR.

## 1. Introduction

Kidney transplantation is considered the best treatment for end-stage kidney disease (ESKD) by offering better long-term outcomes and quality of life (QoL) compared with dialysis treatment [1,2,3]. However, health-related QoL of kidney transplant recipients (KTR) is still lower than the general population and fatigue is an important contributor to this impaired QoL of KTR [4,5,6,7]. Reducing symptoms and complications of treatment, such as fatigue, as well as coping with its consequences for everyday life belongs to patients’ research priorities [8].

In advanced kidney disease, fatigue has been defined as “extreme and persistent tiredness, weakness or exhaustion—mental, physical or both” and is seen as “a complex, multidimensional and multifactorial phenomenon” [9]. Fatigue is a frequently reported symptom in chronic kidney disease (CKD), with the highest prevalence in dialysis patients (up to 89 percent) [9]. Although fatigue generally improves after kidney transplantation, the prevalence is still higher than in (age-matched) healthy controls and reported in 33–59 percent of KTR [4,5,7,10,11].

Despite the high prevalence of severe fatigue and its consequences for daily functioning and QoL of KTR, there are no therapeutic intervention studies that are primarily focused on improvement of fatigue. This is probably due to the fact that the underlying mechanisms of fatigue are not well understood, as only a few studies have addressed this issue [6]. Observations from these cross-sectional studies show that depressive symptoms, poor sleep quality, increased perception of exertion, chronic inflammation, obesity and protein energy wasting (PEW) are associated with fatigue in KTR [5,7,10,12,13]. The role of nutrition as a potential modifiable factor of fatigue in KTR has not been investigated yet. To design interventions targeting fatigue, there is an urgent need for better understanding of the potential modifiable factors of fatigue.

Nutritional factors as potential determinants of disease-related fatigue have been studied in cancer patients, because complications that could potentially impair nutritional status (e.g., altered taste, nausea, lack of appetite) are often part of the disease and its treatment [14]. In cancer patients undergoing chemotherapy, low protein intake is associated with a more than twofold higher risk of cancer-related fatigue [15]. Previous studies in KTR showed a suboptimal protein intake in a significant proportion of stable KTR, which is associated with lower muscle mass, graft failure and increased mortality rates [16,17]. However, whether protein intake plays a role in fatigue after kidney transplantation has, to our knowledge, not been investigated. Therefore, the primary aim of this study was to investigate whether protein intake is associated with fatigue and QoL after kidney transplantation. The secondary aim was to identify other potentially modifiable determinants of fatigue. The knowledge of the potential modifiable factors of fatigue, such as protein intake, could be used in the design of future intervention studies.

## 2. Materials and Methods

### 2.1. Study Design and Population

For this cross-sectional study, we used data from the TransplantLines cohort and biobank study, a multi-disciplinary prospective cohort study in solid organ transplant recipients of the University Medical Center Groningen (UMCG) (ClinicalTrials.gov Identifier: NCT03272841) [18]. All measurements took place during a single study visit of each of the participants, so the outcomes (fatigue and QoL) and exposure (protein intake) were assessed at the same time point for each participant. All participants are unique individuals, which was verified by checking their unique research identification code. None of the KTRs were included twice because of re-transplantation.

For this study, we included adult (≥18 years old) stable outpatient KTR, which was defined as having a functioning graft ≥ 1 year after transplantation without known or apparent systemic illnesses (i.e., malignancies, opportunistic infections). We included KTR at a scheduled study visit between June 2015 and October 2019.

Of a total of 2049 outpatient KTR visiting the outpatient clinic at least once yearly, 1034 KTR had been invited for study participation at the time of closure of the database for this study, of which 812 (78.5%) had signed informed consent and finalized the study visit. We excluded all participants with missing data on fatigue or protein intake (*n* = 88), resulting in 730 participants eligible for analyses. No significant differences were found between the included participants and non-participants with respect to sex (58% male in participants versus 57% male in non-participants, *p* = 0.90), age (56 ± 13 years in participants versus 56 ± 14 years in non-participants, *p* = 0.87), estimated glomerular filtration rate (eGFR) (51.2 ± 17.9 mL/min/1.73 m^2^ in participants versus 50.5 ± 19.5 mL/min/1.73 m^2^ in non-participants, *p* = 0.42) and Body Mass Index (BMI) (27.2 ± 4.7 kg/m^2^ in participants versus 26.9 ± 8.2 kg/m^2^ in non-participants, *p* = 0.23). Of 219 KTR we used the data from the 1-year study visit and of 511 KTR we used the data of the study visit at more than one year after transplantation. With regard to the distribution of the study visits across the study period, 27 study visits were performed in 2015, 180 in 2016, 200 in 2017, 230 in 2018 and 93 in 2019.

The study protocol was approved by the institutional review board (IRB Identifier: METC 2014/077) of the UMCG and all study procedures were performed in accordance with the Declaration of Helsinki and the Declaration of Istanbul.

### 2.2. Assessment of Protein Intake

All subjects were instructed to collect a 24-h urine sample according to a strict protocol at the day before their scheduled study visit at the outpatient clinic. Protein intake in grams per day (g/d) was calculated from 24-h urinary urea nitrogen (UUN) using the Maroni formula: ([UUN (g/24-h) + 0.031 × body weight (kg) ] × 6.25) + urinary protein excretion (g/24-h) [19]. For the sensitivity analysis with protein intake expressed as gram per kilogram per day (g/kg/d), we corrected for potential bias arising from underweight and obesity: a BMI < 20 kg/m^2^ was adjusted to 20 kg/m^2^ and a BMI ≥ 30 kg/m^2^ was adjusted to 27.5 kg/m^2^ [20].

### 2.3. Assessment of Fatigue

Fatigue was assessed by the Checklist Individual Strength (CIS) questionnaire as part of a study visit at the outpatient clinic of the UMCG [21]. This study visit took place at the same time of the blood and urine samples collection. The questionnaire was sent out digitally or by post approximately 2–4 weeks prior to the study visit and participants were asked to fill out the questionnaire at home. The CIS questionnaire enquires about four different dimensions of fatigue: fatigue severity, reduced concentration, reduced motivation and reduced physical activity. It consists of 20 statements for which the participant has to indicate on a 7-point Likert-scale to what extent the statement applies to the participant. The CIS-questionnaires provides a total score, as well as scores on the four subscales. A score ≥ 35 on the subscale of fatigue severity indicates severe fatigue [22,23]. The CIS-questionnaire is well-validated and frequently used in research in patients with various illnesses [24,25,26].

### 2.4. Assessment of Quality of Life

The RAND-36 Health Survey (RAND-36) was used to evaluate QoL and was completed by the participants concomitant with the CIS-questionnaire at the same study visit. The RAND-36 consists of 36 closed-ended questions that measure QoL in 8 health domain subscales (physical functioning, role limitations due to physical health, role limitations due to emotional problems, vitality, emotional well-being, social functioning, pain, general health) [27]. Based on a RAND scoring algorithm, the physical composite summary score (PCS) and mental composite summary score (MCS) were calculated. The PCS consists of items from physical functioning, role limitations due to physical health, pain, vitality, general health and social functioning. The MCS consists of items from role limitations due to emotional problems, emotional well-being, general health, vitality and social functioning.

### 2.5. Assessment of Covariates

Data of the covariates were collected during the same study visit, except for the information on primary kidney disease, dialysis treatment and donor organ specifics. These data were retrieved separately from the UMCG Kidney Transplant Database. Blood samples were collected after an 8–12-h fasting period prior to the study visit. Laboratory parameters were measured using stand laboratory procedures. BMI was calculated as weight in kilograms divided by height in squared meter. Blood pressure was measured using an automatic device (Philips Suresign VS2+, Andover, Massachusetts, USA) in a seated position. The Patient-Generated Subjective Global Assessment (PGSGA) was used for assessment of nutritional status and according to the PG-SGA Global Assessment Category patients were classified as well-nourished (stage A), moderately malnourished (stage B) or severely malnourished (stage C) [28]. Diabetes was diagnosed when at least one of the following criteria was met: (1) symptoms of diabetes (e.g., polyuria, polydipsia, unexplained weight loss) plus a non-fasting plasma glucose concentration ≥ 11.1 mmol/L (200 mg/dL), (2) fasting plasma glucose ≥ 7.0 mmol/L (126 mg/dL), (3) plasma Hemoglobin A1c ≥ 6.5% (48 mmol/mol) or (4) use of antidiabetic medication [29,30]. eGFR was calculated using the serum creatinine-based CKD-EPI algorithm [31]. Proteinuria was defined as urinary protein excretion ≥ 0.5 g/24-h. Data of smoking status, alcohol use, level of education and employment status were obtained from questionnaires.

### 2.6. Statistical Analyses

Normally distributed data are presented as means ± standard deviations, non-normally distributed data are presented as median [interquartile range] and categorical data are presented in numbers (percentages). For testing differences in sex, age, eGFR and BMI of the included participants of this study versus the non-participants (Section 2.1), the independent t-test was used for normally distributed data and the chi-square test for categorical data. Based on the CIS subscale score of subjective fatigue subjects were divided into three groups. Patients with a CIS-score < 20 were considered to have no to mild fatigue, 20–34 were considered to have moderate fatigue and those with a score ≥ 35 were considered to have severe fatigue [22,23]. Differences in baseline characteristics across categories of fatigue severity were tested by ANOVA for normally distributed data, Kruskal–Wallis test for non-normally distributed data and chi-square test for categorical data. To investigate the association of protein intake with severity of fatigue we performed multinomial logistic regression analyses, with no to mild fatigue, moderate fatigue and severe fatigue as categories of the dependent variable. We analyzed protein intake as a continuous variable expressed in g/d. We refrained from the common use to express protein in g/kg/d to avoid introduction of a potential systematic error as a consequence of the high prevalence of overweight and obesity, as well as abnormalities in body composition in our cohort (e.g., lower muscle mass). To be able to relate to data in the literature, we performed a sensitivity analyses for protein intake expressed in g/kg/day, with adjustments for underweight and obesity as described before. In multivariable multinomial logistic regression analyses, we cumulatively adjusted the association of protein intake with categories of fatigue, i.e., no to mild fatigue, moderate fatigue and severe fatigue for potential confounders, including, age, sex and BMI (model 1), eGFR, proteinuria, primary kidney disease (model 2), time after transplantation, pre-emptive transplantation, living kidney donor, mycophenolic acid, cyclosporin use and a history of combined organ transplantation (model 3), diabetes, systolic blood pressure, cholesterol, and statin use (model 4), hemoglobin-, ferritin-, vitamin B12-, albumin and C-reactive protein (CRP) concentrations (model 5) and smoking status, alcohol use and level of education (model 6). Similarly, the association of protein intake with the PCS and MCS of QoL was analyzed using univariable and multivariable linear regression analyses. Statistical analyses were performed using SPSS version 23.0 (IBM Corp., Armonk, NY) and R-Studio version 1.2.1 (R Foundation for Statistical Computing, Vienna, Austria). In all analyses, a *p*-value < 0.05 was considered statistically significant.

## 3. Results

### 3.1. Baseline Characteristics of Total Population and Across Categories of Fatigue

We included 730 KTR with a study visit at a median of 4.1 [IQR 1.0–11.0] years after transplantation. Subjects were 58 [48–65] years of age, 57% were male, and mean eGFR was 51.2 ± 17.9 mL/min/1.73 m^2^. A combined organ transplantation was performed in 20 KTR, of which 17 received a combined kidney–pancreas transplantation and 3 received a combined kidney–liver transplantation. Moderate and severe fatigue was present in 254 (35%) and 245 (34%) of KTR, respectively. Daily protein intake of the total study population was 82.2 ± 21.3 g per day (g/d), corresponding to 1.07 ± 0.25 g/kg/d. KTR with severe fatigue had a lower daily protein intake (78.9 ± 19.3 g/d), compared with KTR without fatigue or with moderate fatigue (86.4 ± 14.7 g/d and 81.4 ± 21.9 g/d respectively). Baseline characteristics of the total study population and across categories of fatigue are shown in Table 1.

Compared with KTR without fatigue or with moderate fatigue, KTR with severe fatigue had a higher BMI (*p* = 0.01), lower 24-h creatinine excretion (*p* = 0.002) and eGFR (*p* = 0.01) as well as lower levels of albumin (*p* < 0.001) and hemoglobin (*p* = 0.008). They were also more often diagnosed with diabetes (*p* = 0.001) and more often classified as moderately or severely malnourished by the PG-SGA (*p* < 0.001). The employment status was significantly different across the groups of fatigue severity, as KTR with severe fatigue reported lower rates of paid employment and were more often medically unfit for work. Regarding the kidney transplant characteristics, KTR with severe fatigue received a kidney transplant longer time ago (*p* < 0.001), with less frequently use of mycophenolic acid (*p* = 0.002) and more often use of cyclosporine (*p* = 0.01). KTR with severe fatigue also less often had a pre-emptive transplantation (*p* = 0.001), received a kidney from a living donor less often (*p* = 0.008), more often had proteinuria (*p* = 0.01) and more often had a history of combined organ transplantation (*p* = 0.02).

### 3.2. Fatigue

In univariable multinomial logistic regression analyses, protein intake was significantly associated with risk of moderate fatigue (OR 0.89 per 10 g/d increment; 95%CI 0.83–0.98, *p* = 0.01) and severe fatigue (OR 0.85; 95%CI 0.78–0.92, *p* < 0.001). This association with severe fatigue remained significant in multivariable regression analyses after cumulative adjustment for potential confounders, including age, sex, BMI (model 1), kidney function parameters (model 2), transplantation specific characteristics (model 3), cardiovascular risk factors (model 4), hemoglobin, ferritin, vitamin B12, albumin and CRP levels (model 5), and lastly for lifestyle parameters and education (model 6) as shown in Table 2.

In the final cumulative model (model 6), higher BMI, a history of dialysis, glomerulonephritis as primary kidney disease and a history of combined organ transplantation were also associated with severe fatigue (Appendix A). Age-, sex- and BMI-adjusted associations between protein intake with moderate and severe fatigue are visualized by restricted cubic splines in Figure 1.

### 3.3. Quality of Life

Data on QoL were available of 693 (95%) subjects. In subjects with CIS ≥ 35, both the PCS and MCS were significantly lower compared with subjects with a CIS score < 20 or between 20 and 35. In univariable linear regression analyses, protein intake (g/d) was associated with PCS (β 0.74 per 10 g/d increment; 95%CI 0.39–1.09, *p* < 0.001) and MCS (β 0.36 per 10 g/d increment; 95%CI 0.06–0.66, *p* = 0.02). As shown in Table 3, the association between protein intake and PCS remained significant after adjustment for several potential confounders. For MCS, the significant association was lost in multivariable linear regression analysis.

### 3.4. Sensitivity Analyses

In the sensitivity analyses with protein intake in g/kg/d, we observed that protein intake was also associated with a lower risk of moderate (OR 0.91 per 0.1 g/kg increment; 95%CI 0.85–0.98, *p* = 0.01) and severe fatigue (OR 0.85 per 0.1 g/kg increment; 95%CI 0.79–0.92, *p* < 0.001). The association with severe fatigue remained significant in multivariable regression analyses after adjustment for potential confounders (Appendix A). For QoL, protein intake in g/kg/d was significantly associated with PCS (β 0.62 per 0.1 g/kg/d increment; 95%CI 0.32–0.91, *p* < 0.001), but not with MCS (β 0.19 per 0.1 g/kg/d increment; 95%CI −0.07–0.44, *p* = 0.16) in univariable analyses. These findings remained materially unchanged after adjustment for the several potential confounders (Appendix A).

## 4. Discussion

In this large observational cohort study, we investigated the association of protein intake with fatigue severity and QoL in stable outpatient KTR. To our knowledge, this is the first study that shows that higher protein intake is independently associated with lower risk of moderate and severe fatigue and better PCS of QoL after kidney transplantation.

Given the high prevalence of fatigue in KTR and its detrimental effects on daily functioning and QoL, it is of paramount importance to identify potential modifiable factors of fatigue that could be targeted by an intervention. In this patient cohort, severe fatigue was present in 34% percent of the KTR. These findings are in line with the fatigue prevalence of 33–59% in previous studies in KTR using the CIS-questionnaire [4,5,7,10,11]. Besides the association with protein intake, higher BMI, a history of dialysis, glomerulonephritis as primary kidney disease, and a history of combined organ transplantation were also independently associated with severe fatigue in this cohort. Only few other studies identified several demographic, psychological, physical and disease-related factors that are associated with fatigue after kidney transplantation. Somewhat surprisingly, the role of nutritional factors in fatigue after kidney transplantation has not been explored previously. Moreover, diagnostic and therapeutic considerations, or practical approaches, of fatigue are currently not part of CKD or kidney transplantation guidelines, resulting in limited practical support for clinicians. In clinical practice, therefore, nutritional factors, and possibly also other modifiable factors, may be overlooked in the diagnostic work-up of fatigue after kidney transplantation. Our finding that protein intake is independently associated with severe fatigue and QoL in KTR highlights the need for involvement of nutrition in the (multidimensional) assessment and possibly treatment of fatigue. Our data suggest that nutritional assessment could incorporate 24-h urine samples for objective measurement of protein intake. Furthermore, attention should be paid to the (concomitant) presence of malnutrition and overweight or obesity. In our study the PG-SGA was used as malnutrition tool, which indeed showed a higher rate of malnutrition in KTR with severe fatigue.

Only few studies explored the relationship between protein intake with either fatigue or QoL in chronic disease, and all have been performed in cancer patients. Our results are consistent with findings by Stobäus et al. who also observed protein intake was associated with cancer-related fatigue in advanced cancer patients that underwent chemotherapy [15]. Furthermore, in hospitalized cancer patients low protein, and not caloric intake, was associated with poorer perception of physical functioning and fatigue [32].

The association between protein intake with severe fatigue and PCS of QoL could be explained by several mechanisms. First, a higher protein intake may contribute to the preservation of muscle mass and muscle strength, thereby contributing to better physical functioning and, as a consequence, less fatigue. In KTR low protein intake has been associated with low muscle mass, which explained the association between lower protein intake and a higher mortality risk [17]. Our study showed that lower protein intake was associated with fatigue and lower PCS of QoL, as well as lower creatinine excretion, a marker of muscle mass. Progressive loss of muscle mass and muscle strength is a key component of frailty—a state of increased vulnerability which is increasingly recognized in KTR and associated with worse outcomes [33,34]. It recently has been hypothesized that creatine becomes an essential amino acid in kidney disease, due to insufficient endogenous synthesis of creatine caused by kidney impairment [35]. Insufficient intake may lead to a relative creatine deficiency, contributing to muscle weakness, low muscle mass, impaired cognition and fatigue in patients with kidney function impairment. This could be a mechanism underlying the vulnerability of kidney patients to a relatively low protein intake.

Second, a higher protein intake may serve as a preventive measure for malnutrition, by ensuring an adequate intake relative to the possibly increased protein demand in KTR. Several transplant-specific factors predispose KTR for development of malnutrition, including infectious complications, rejection episodes, insulin resistance, use of immunosuppressive medication and the immune response to the graft [36,37,38,39], which function as catabolic stimuli. Consequently, the presence of this (hyper) catabolic state in KTR leads to higher protein and energy requirements. In previous studies, the presence of malnutrition has been linked to lower QoL and fatigue as well as a higher mortality in KTR, underscoring the importance of maintaining an adequate nutritional status [12,40]. In a previous KTR cohort from our own center, we found excess mortality below a protein intake of ~1.1 g/kg/d [16]. Despite the higher protein requirements after kidney transplantation, a suboptimal protein intake was observed in more than half of our cohort. This may be a consequence of continuation of a previous protein restriction at time of ESKD. While such dietary restrictions are no longer required after transplantation and recovery of kidney function, this dietary transition may function as a barrier to resume a diet with sufficient protein [41].

Third, it is increasingly recognized that disease-related fatigue is related to the presence of chronic inflammation. Among several different nutrients and dietary pattern, it is suggested that specific proteins (such as soy protein) may have anti-inflammatory properties [42]. However, there is currently lack of evidence that supplementation of specific nutrients (amino acids) leads to significant improvement of fatigue. Finally, the pathophysiologic causes of fatigue, e.g., chronic inflammation, may also lead to both fatigue and a poor nutritional intake.

One of the strengths of this study is the use of 24-h UUN to assess protein intake in an objective way. This method is based on the concept of nitrogen balance, where the net urea production (the principle end product of amino acid degradation) parallels the protein intake under stable circumstances. In steady state, 24-h UUN is equal to the net urea production, allowing to calculate protein intake from 24-h urinary collections [43]. This method based on nitrogen balance is considered the gold standard for protein intake assessment [44,45]. The method is validated in CKD-patients [43] and regarded to be more reliable than the use of food diaries or questionnaires to calculate protein intake, as these information sources based on self-report are prone to bias and errors [46]. Another strength of this study is the large number of stable KTR that participate in this study. This study also has several limitations. First, within the cross-sectional observational study design, a causal relationship could not be determined. A lower protein intake could also be a consequence of fatigue. Low energy levels, for instance, affects daily functioning that includes grocery shopping and preparation of meals and this may lead to poorer nutritional intake. Second, the data collection for this cross-sectional analysis took place in a time span of > 4 years. It cannot be excluded that this impacted the results by changes in transplant care or other, external factors within this time frame. It should, however, be noted that in this time frame there were neither changes in the immunosuppressive regimen nor in supportive care, including, e.g., the antiviral and antimicrobial prophylaxis. In addition, other aspects of kidney transplant care and the health care system also remained materially unchanged. Third, we were not able to correct our finding for total energy intake, and could not exclude that the low protein intake was accompanied by a low caloric intake, that could also contribute to fatigue. Finally, as this is a single center study, we were not able to ascertain whether the KTR of our study population are representative of all KTR of the Netherlands. Similarly, we cannot ascertain whether our results can be extrapolated to KTR in general.

This study has several implications for clinical practice. First, our findings underscore that the diagnostic work-up and therapeutic strategies for fatigue should incorporate proper nutritional assessment and counseling. The optimal protein requirements for stable KTR are not well defined; it is assumed that nutritional recommendations for CKD patients are also sufficient after kidney transplantation, with a tendency to limit protein intake to ~0.8 g/kg/d. However, this study as well as previous studies suggest that KTR may benefit from a higher protein intake for both their wellbeing as well as long-term patient and kidney outcomes [16,17]. Results from these observational studies show that the risk of both fatigue and the risk of graft failure and mortality increases below a protein intake of ~1.1 g/kg/d. Based on these observations the optimal protein intake for KTR is in line with recommendations for dialysis patients [47] and for elderly to maintain optimal muscle function [48]. Although we cannot advice on the upper limit for protein intake for KTR based on our findings, survival of dialysis patients did not further improve above a protein intake of 1.4 g/kg/d [49]. The Kidney Disease Improving Global Outcomes (KDIGO) guideline for CKD patients recommends to avoid excessive protein intake (>1.3 g/kg/d) in non-diabetic CKD-patients with an eGFR > 30 mL/min/1.73 m^2^ [50]. Currently, no specific recommendations are available for KTR, but an ongoing dietary intervention study in 120 KTR, aiming to evaluate the effect of a high-protein (1.3–1.4 g/kg/d) low-glycemic-index diet on prevention of post-transplant weight gain, may provide more insight in the optimal protein intake after kidney transplantation [51]. With the high prevalence of overweight, obesity and its cardio-metabolic derangement, as well as the high risk of malnutrition, tailored dietary advice in collaboration with a renal dietician is of paramount importance. Furthermore, future dietary intervention studies are required to assess if a higher protein intake indeed contributes to improvement of fatigue and QoL. Given the multifactorial nature of fatigue in KTR, this should preferably be part of a (individualized) multimodal therapeutic intervention that incorporates assessment and management of both physiological factors (e.g., anemia, vitamin deficiencies) and of nutritional and other lifestyle factors (e.g., obesity, physical inactivity) as well as the psychosocial context [9].

## 5. Conclusions

In conclusion, in this study we show that higher protein intake is independently associated with a lower risk of moderate and severe fatigue and better QoL in KTR. These findings indicate that assessment of intake and nutritional status, which could be complemented by 24-h urine measurements, should be incorporated in the diagnostic work-up of fatigue after kidney transplantation. Future intervention studies are needed to assess the efficacy and safety of higher protein intake for improvement of fatigue and QoL in KTR.

## Figures and Tables

**Figure 1 nutrients-12-02451-f001:**
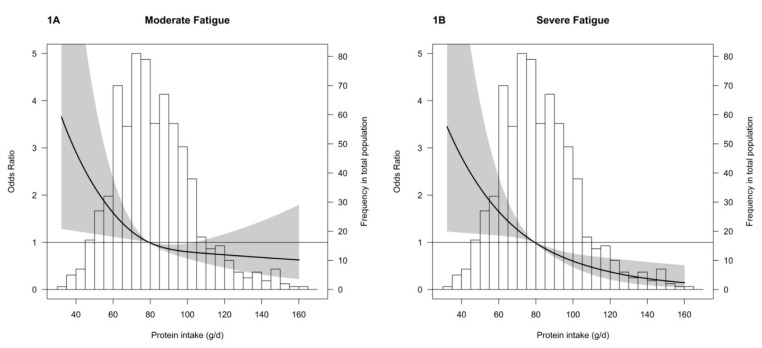
Age-, sex- and BMI-adjusted associations of protein intake with moderate and severe fatigue. Legend Figure 1: Restricted cubic splines showing the association of protein intake (in grams per day) with risk of moderate fatigue (**A**) and severe fatigue (**B**) after adjustment for age, sex and BMI. The black line represents the odds ratio estimate and the grey represent the 95% confidence interval.

**Table 1 nutrients-12-02451-t001:** Baseline characteristics of 730 KTR and differences according to categories of fatigue.

	CIS				
	Total Population	<20	20–35	≥35	*p*-Value
Number of Subjects	730 (100)	231 (31)	254 (35)	245 (34)	-
24-h Urea excretion, mmol	377 ± 114	403 ± 119	373 ± 118	356 ± 101	<0.001
Daily protein intake, g/day	82.2 ± 21.3	86.4 ± 14.7	81.4 ± 21.9	78.9 ± 19.3	<0.001
**Quality of Life**					
PCS	45.0 ± 10.0	52.0 ± 5.3	45.6± 8.6	37.6 ± 9.9	<0.001
MCS	50.6 ± 8.6	55.0 ± 4.9	50.9 ± 7.6	46.0 ± 10.2	<0.001
**Demographics**					
Sex, male	423 (57)	147 (64)	142 (56)	134 (55)	0.10
Age, years	58 [48–65]	58 [46–65]	59 [48–66]	57 [49–65]	0.53
Education					0.30
Low	265 (36)	73 (32)	104 (41)	88 (36)	
Intermediate	239 (33)	83 (36)	72 (28)	84 (34)	
High	175 (24)	61 (26)	61 (24)	53 (22)	
Unknown/PNA	51 (7)	14 (6)	17 (7)	20 (8)	
Employment					<0.001
Paid employment	274 (38)	115 (50)	91 (36)	68 (28)	
Medically unfit for work	143 (20)	18 (8)	47 (18)	78 (32)	
Unemployed	74 (10)	21 (9)	32 (13)	21 (8)	
Retired	186 (25)	61 (26)	67 (26)	58 (24)	
Unknown	53 (7)	16 (7)	17 (7)	20 (8)	
**Kidney Transplant Characteristics**					
Primary kidney disease					0.29
Glomerulonephritis	166 (23)	48 (21)	55 (22)	63 (25)	
Interstitial Nephritis	67 (9)	18 (8)	25 (10)	24 (10)	
Cystic Kidney Disease	147 (20)	55 (24)	48 (19)	44 (18)	
Other congenital and hereditary kidney disease	44 (6)	14 (6)	12 (5)	18 (7)	
Renal vascular disease	93 (13)	35 (15)	37 (14)	21 (9)	
Diabetes Mellitus	44 (6)	9 (4)	14 (5)	21 (9)	
Other multisystem diseases	31 (4)	8 (3)	11 (4)	12 (5)	
Other	17 (2)	6 (2)	4 (2)	7 (3)	
Unknown	121 (17)	38 (17)	48 (19)	35 (14)	
Time since Tx, years	4.0 [1.0–11.0]	2.0 [1.0–7.9]	4.0 [1.0–10.8]	6.4 [1.3–13.0]	<0.001
History of combined organ Tx	20 (3)	1 (0.4)	8 (3)	11 (5)	0.02
eGFR, mL/min × 1.73 m^2^	51.2 ± 17.9	53.1 ± 17.7	51.9 ± 17.0	48.6 ± 18.8	0.01
Proteinuria	110 (15)	34 (15)	27 (11)	49 (20)	0.01
Pre-emptive Tx	283 (39)	110 (48)	98 (39)	75 (31)	0.001
Donor sex, male	382 (52)	114 (50)	130 (52)	138 (59)	0.19
Donor age, years	52 [43–60]	54 [45–62]	52 [43–60]	49 [37–58]	0.003
Living donor	412 (56)	143 (62)	150 (59)	119 (49)	0.008
**Body Composition and Nutritional Status**					
Weight, kg	81.8 ± 16.3	80.2 ± 14.7	81.4 ± 15.0	83.7 ± 18.6	0.06
Height, cm	173 ± 10	173 ± 11	172 ± 9	173 ± 10	0.41
BMI, kg/m^2^	27.3 ± 4.7	26.6 ± 4.3	27.4 ± 4.5	27.9 ± 5.3	0.01
24-h CER, mmol	12.2 ± 3.8	12.9 ± 3.9	12.1 ± 3.7	11.7 ± 3.7	0.002
PGSGA stage B or C	47 (8)	4 (2)	11 (6)	32 (17)	<0.001
**Cardio-Metabolic Parameters**					
SBP, mm Hg	135 ± 16	135 ± 15	134 ± 16	134 ± 16	0.64
DBP, mm Hg	79 ± 10	79 ± 10	78 ± 10	78 ± 11	0.42
Use of antihypertensive drugs	582 (80)	180 (78)	201 (79)	201 (82)	0.51
Total cholesterol, mmol/L	4.7 ± 1.0	4.7 ± 1.0	4.6 ± 0.9	4.8 ± 1.1	0.07
LDL-cholesterol, mmol/L	2.9 ± 0.9	2.9 ± 0.8	2.8 ± 0.8	2.9 ± 1.0	0.15
Statin use	425 (58)	134 (58)	157 (62)	134 (55)	0.27
Diabetes	208 (29)	48 (21)	70 (34)	90 (37)	0.001
HbA1c, mmol/mol	5.8 [5.4–6.3]	5.7 [5.4–6.1]	5.8 [5.4–6.3]	5.9 [5.6–6.5]	0.003
**Inflammatory and Hematological Parameters**					
Albumin, g/L	43.5 ± 3.0	44.2 ± 2.7	43.6 ± 2.8	42.9 ± 3.4	<0.001
CRP, mg/L	2.0 [0.8–4.9]	1.6 [0.7–3.7]	2.4 [0.8–5.0]	2.5 [0.8–6.0]	0.02
Hemoglobin, mmol/L	8.3 ± 1.1	8.4 ± 1.2	8.4 ± 1.0	8.1 ± 1.1	0.008
Iron, µg/dL	13.8 ± 5.6	14.3 ± 5.1	13.6 ± 5.5	13.6 ± 6.0	0.34
Ferritin, µg/L	89 [41–189]	90 [41–193]	88 [39–198]	89 [44–178]	0.9
Vitamin B12, pmol/L	289 [219–391]	288 [226–375]	291 [221–391]	288 [214–411]	0.84
**Immunosuppressive Drugs**					
Tacrolimus	483 (66)	160 (69)	168 (66)	155 (63)	0.39
Cyclosporin	109 (15)	29 (13)	30 (12)	50 (20)	0.01
Mycophenolic acid	545 (75)	181 (78)	201 (79)	163 (67)	0.002
Azathioprine	78 (11)	21 (9)	28 (11)	29 (12)	0.61
Prednisolone	711 (97)	225 (97)	247 (97)	293 (98)	0.9
**Lifestyle Parameters**					
Smoking status					0.36
Yes	83 (11)	23 (10)	25 (10)	35 (14)	
No	617 (85)	198 (86)	221 (87)	198 (81)	
Unknown	30 (4)	10 (4)	8 (3)	12 (5)	
Alcohol use					0.37
Yes	364 (50)	124 (54)	130 (52)	110 (45)	
No	229 (31)	68 (29)	75 (39)	86 (35)	
Unknown	137 (19)	39 (17)	49 (19)	49 (20)	

Note: Data are presented as mean ± SD, number (%) or median [IQR]. Abbreviations: 24-h CER: 24 h creatinine excretion rate; BMI: body mass index; CIS: Checklist Individual Strength; CRP: C-reactive protein; DBP: diastolic blood pressure; eGFR: estimated glomerular filtration rate; HbA1c: Hemoglobin A1c; LDL: low-density-lipoprotein; QoL: quality of life, measured by RAND-36; PGSGA: Patient-Generated Subjective Global Assessment—stage B and C corresponds with moderate and severe malnutrition respectively; SBP: systolic blood pressure; Tx: transplantation.

**Table 2 nutrients-12-02451-t002:** Association of protein intake (per 10 g/d increment) with fatigue in KTR.

	No-Mild Fatigue		Moderate Fatigue		Severe Fatigue	
CIS	<20		20–34		≥35	
	OR (95% CI)	*p*-Value	OR (95% CI)	*p*-Value	OR (95% CI)	*p*-Value
Crude	Reference	(-)	0.89 (0.83–0.98)	0.01	0.85 (0.78–0.92)	<0.001
Model 1	Reference	(-)	0.87 (0.79–0.96)	0.005	0.79 (0.72–0.88)	<0.001
Model 2	Reference	(-)	0.86 (0.78–0.95)	0.004	0.78 (0.70–0.87)	<0.001
Model 3	Reference	(-)	0.87 (0.79–0.96)	0.006	0.80 (0.72–0.90)	<0.001
Model 4	Reference	(-)	0.88 (0.79–0.97)	0.01	0.80 (0.72–0.90)	<0.001
Model 5	Reference	(-)	0.90 (0.80–1.00)	0.06	0.80 (0.70–0.90)	<0.001
Model 6	Reference	(-)	0.89 (0.80–1.00)	0.06	0.80 (0.71–0.91)	<0.001

Abbreviations: 95% CI: 95% confidence interval; CIS: Checklist Individual Strength; g/kg/d: grams per kilogram per day; OR: Odds Ratio. Model 1: adjusted for age, sex, BMI. Model 2: adjusted for model 1 variables plus eGFR, proteinuria and primary kidney disease. Model 3: adjusted for model 2 variables plus time after transplantation, pre-emptive transplantation, living kidney donor, mycophenolic acid use, cyclosporine use and a history of combined organ transplantation. Model 4: adjusted for model 3 variables plus diabetes, systolic blood pressure and cholesterol and statin use. Model 5: adjusted for model 4 variables plus hemoglobin, ferritin, vitamin B12, albumin, C-reactive protein. Model 6: adjusted for model 5 variables plus smoking status, alcohol use and level of education.

**Table 3 nutrients-12-02451-t003:** Association of protein intake (per 10 g/d increment) with Quality of Life.

	PCS		MCS	
	β (95% CI)	*p*-Value	β (95% CI)	*p*-Value
Crude	0.74 (0.39–1.09)	<0.001	0.36 (0.06–0.66)	0.02
Model 1	0.97 (0.60–1.34)	<0.001	0.25 (−0.09–0.59)	0.15
Model 2	0.97 (0.60–1.34)	<0.001	0.30 (−0.04–0.64)	0.09
Model 3	0.83 (0.46–1.21)	<0.001	0.25 (−0.10–0.59)	0.16
Model 4	0.74 (0.36–1.12)	<0.001	0.23 (−0.12–0.58)	0.20
Model 5	0.65 (0.24–1.06)	0.002	0.26 (−0.12–0.64)	0.19
Model 6	0.64 (0.23–1.05)	0.002	0.26 (−0.12–0.65)	0.18

Abbreviations: 95% CI: 95% confidence interval; g/d: grams per day; g/kg/day: grams per kilogram per day; MCS: Mental component summary score; PCS: Physical component summary score. Model 1: adjusted for age, sex, BMI. Model 2: adjusted for model 1 variables plus eGFR, proteinuria and primary kidney disease. Model 3: adjusted for model 2 variables plus time after transplantation, pre-emptive transplantation, living kidney donor, mycophenolic acid use, cyclosporin use and history of combined organ transplantation. Model 4: adjusted for model 3 variables plus diabetes, systolic blood pressure and cholesterol and statin use. Model 5: adjusted for model 4 variables plus hemoglobin, ferritin, vitamin B12, albumin, C-reactive protein. Model 6: adjusted for model 5 variables plus smoking status, alcohol use and level of education.

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
