# Peer review of "Protein Intake, Fatigue and Quality of Life in Stable Outpatient Kidney Transplant Recipients"

_nutrients, 2020, doi:10.3390/nu12082451_

Round 1
Reviewer 1 Report
This study focused on the association between protein intake and fatigue, QoL in kidney transplant recipients. The study design is clear and the results are also presented well. I have some comments to improve this manuscript:
- Statistical methods/Results: the author used multinominal logistic regression to show the association between risk factors and fatigue. I am wondering the results of cumulative logistic regression to model the accumulative risk of fatigue severity?
- The assessment of Protein intake was calculated 24-hrs urine sample. The author stated that it is a more objective measurement than food diaries or questionnaires. Would the author address more information regarding the "golden standard of the protein intake evaluation"? This would enhance the value of 24-hrs urine samples.
- In this study, the author suggested KRT may benefit from a higher protein intake (~1.1h/kg/d), which similar to previous studies. I think this is an important issue. I would suggest the author extend some discussions on this point. Meanwhile, is there any upper limit for KRT?
- P3, assessment of the quality of life: please cite the reference
- Finally, I did not see the supplementary materials. Would you please check it?
Author Response
Dear Editor,
We thank the reviewers for the valuable comments and the opportunity to submit a revised version of our manuscript to be considered for publication. We have changed the manuscript according to the suggestions of all reviewers, to strengthen our manuscript. A detailed, itemized response to all comments can be found in this document. Please find enclosed our revised manuscript with ID nutrients-874426, entitled “Protein Intake, Fatigue and Quality of Life in Stable Outpatient Kidney Transplant Recipients.”
Sincerely,
On behalf of all the authors,
Karin Boslooper-Meulenbelt
This study focused on the association between protein intake and fatigue, QoL in kidney transplant recipients. The study design is clear and the results are also presented well. I have some comments to improve this manuscript:
- Statistical methods/Results: the author used multinominal logistic regression to show the association between risk factors and fatigue. I am wondering the results of cumulative logistic regression to model the accumulative risk of fatigue severity?
Response: we thank the reviewer for this comment to provide more insight in the cumulative logistic regression model to model the accumulative risk of fatigue severity. The cumulative regression model was provided as supplementary Table S1 as part of the original manuscript. We provide our sincere apologies that this document was not visible at the time of this review (as mentioned at comment #5 of this reviewer). The comment of the reviewer also made us realize that it can be clearer described in the Results section and the title of the supplementary document that we here show the results of the cumulative logistic regression model. To accommodate the comment of the reviewer, we now added the supplementary documents enclosed as separate file of the revised manuscript.
Based on this comment the we also adapted the sentence in which the model was mentioned in the manuscript. This sentence now reads as follows: ‘In the final cumulative model (model 6), higher BMI, a history of dialysis, glomerulonephritis as primary kidney disease and a history of combined organ transplantation were also associated with severe fatigue (Supplementary table S1).’ (see Chapter Results, line 243-246 in the manuscript with tracked changes).
Additionally, the title of Supplementary Table S1 is changed to ‘Table S1. Final cumulative model of multinomial regression analyses of protein intake (per 10g/d increment) with fatigue in KTR’.
To further accommodate the comment of the reviewer, we also made more clear that we used three categories of the dependent variable fatigue in the multinomial logistic regression to show the accumulative risk of fatigue severity, i.e. no to mild fatigue, moderate fatigue and severe fatigue. Accordingly, the following sentences were adapted in the revised version of the manuscript:
‘To investigate the association of protein intake with severity of fatigue we performed multinomial logistic regression analyses, with no to mild fatigue, moderate fatigue and severe fatigue as categories of the dependent variable’ (Chapter Materials and Methods, paragraph 2.6, line 171 – 173 in the manuscript with tracked changes)
‘In multivariable multinomial logistic regression analyses, we cumulatively adjusted the association of protein intake with categories of fatigue, i.e. no to mild fatigue, moderate fatigue and severe fatigue for potential confounders, including, age, sex and BMI (model 1), eGFR, proteinuria, primary kidney disease (model 2), time after transplantation, pre-emptive transplantation, living kidney donor, mycophenolic acid, ciclosporin use and a history of combined organ transplantation (model 3), diabetes, systolic blood pressure, cholesterol, and statin use (model 4), hemoglobin-, ferritin-, vitamin B12-, albumin and C-reactive protein (CRP) concentrations (model 5) and smoking status, alcohol use and level of education (model 6).’ (Chapter Material and Methods, paragraph 2.6, line 178-186 in the manuscript with tracked changes)
- The assessment of Protein intake was calculated 24-hrs urine sample. The author stated that it is a more objective measurement than food diaries or questionnaires. Would the author address more information regarding the "golden standard of the protein intake evaluation"? This would enhance the value of 24-hrs urine samples.
Response: We thank the reviewer for this comment, which gives us the opportunity to provide more in-depth information about the use of 24-hour urinary urea excretion rate to calculate protein intake.
Based on this comment we added extra information on this topic to the discussion
section of the revised version of the manuscript: ‘One of the strengths of this study is the use
of 24-hour UUN to assess protein intake in an objective way. This method is based on the
concept of nitrogen balance, where the net urea production (the principle end product of
amino acid degradation) parallels the protein intake under stable circumstances. In steady
state, 24-hour UUN is equal to the net urea production, allowing to calculate protein intake
from 24-hour urinary collections [43]. This method based on nitrogen balance is considered
the gold standard for protein intake assessment [44,45]. The method is validated in CKD-
patients [43] and regarded to be more reliable than the use of food diaries or questionnaires to
calculate protein intake, as these information sources based on self-report are prone to bias
and errors [46].’ (Chapter Discussion, line 351 – 359 in the manuscript with tracked changes).
Accordingly, we also added three extra references
[43]: Masud, T.; Manatunga, A.; Cotsonis, G.; Mitch, W.E. The precision of estimating
protein intake of patients with chronic renal failure. Kidney Int. 2002,
62, 1750–6, doi:10.1046/j.1523-1755.2002.00606.x.
[44] Mitch, W.E. Dietary protein restriction in patients with chronic renal failure. Kidney Int.
1991, 40, 326–41, doi:10.1038/ki.1991.217.
[45] Matsuda, T.; Kato, H.; Suzuki, H.; Mizugaki, A.; Ezaki, T.; Ogita, F. Within-Day Amino
Acid Intakes and Nitrogen Balance in Male Collegiate Swimmers during the General
Preparation Phase. Nutrients 2018, 10, doi:10.3390/nu10111809.
- In this study, the author suggested KRT may benefit from a higher protein intake (~1.1h/kg/d), which similar to previous studies. I think this is an important issue. I would suggest the author extend some discussions on this point. Meanwhile, is there any upper limit for KRT?
Response: We thank the reviewer for this comment and the opportunity to provide additional information on this topic. Based on this comment, we added extra information on this topic to the discussion section of the revised version of the manuscript:
‘Results from these observational studies show that the risk of both fatigue and the risk of graft failure and mortality increases below a protein intake of ~1.1 g/kg/d. Based on these observations, the optimal protein intake for KTR is in line with recommendations for dialysis patients [47] and for elderly to maintain optimal muscle function [48]. Although we cannot advice on the upper limit for protein intake for KTR based on our findings, survival of dialysis patients did not further improve above a protein intake of 1.4 g/kg/d. [49]. The Kidney Disease Improving Global Outcomes (KDIGO) guideline for CKD patients recommends to avoid excessive protein intake (>1.3 g/kg/d) in non-diabetic CKD-patients with an eGFR >30 mL/min/1.73m2 [50]. Currently, no specific recommendations are available for KTR, but an ongoing dietary intervention study in 120 KTR, aiming to evaluate the effect of a high-protein (1.3-1.4 g/kg/d) low glycemic-index diet on prevention of post-transplant weight gain, may provide more insight in the optimal protein intake after kidney transplantation [51].’ (Chapter Discussion, Line 376-388 in the manuscript with tracked changes)
Accordingly, we also added five extra references:
- Kopple, J.D.; National Kidney Foundation K/DOQI Work Group The National Kidney
Foundation . K/DOQI clinical practice guidelines for dietary protein intake for chronic
Dialysis patients. Am. J. Kidney Dis. 2001, 38, S68 - 73, doi:10.1053/ajkd.2001.27578.
- Deutz, N.E.P.; Bauer, J.M.; Barazzoni, R.; Biolo, G.; Boirie, Y.; Bosy-Westphal, A.;
Cederholm, T.; Cruz Jentoft, A.; Krznariç, Z.; Nair, K.S.; et al. Protein intake and exercise for
optimal muscle function with aging: recommendations from the ESPEN Expert Group. Clin.
Nutr. 2014, 33, 929–36, doi:10.1016/j.clnu.2014.04.007.
- Shinaberger, C.S.; Kilpatrick, R.D.; Regidor, D.L.; McAllister, C.J.; Greenland, S.;
Kopple, J.D.; Kalantar Zadeh, K. Longitudinal associations between dietary protein intake
and survival in hemodialysis patients. Am. J.Kidney Dis. 2006, 48, 37–49,
doi:10.1053/j.ajkd.2006.03.049.
- Stevens, P.E.; Levin, A.; Kidney Disease: Improving Global Outcomes Chronic Kidney
Disease Guideline Development Work Group Members Evaluation and management of
chronic kidney disease: synopsis of the kidney disease: improving global outcomes 2012
clinical practice guideline. Ann. Intern. Med. 2013, 158, 825–30,
doi:10.7326/0003-4819-158-11-201306040-00007.
- Pedrollo, E.F.; Nicoletto, B.B.; Carpes, L.S.; de Freitas, J. de M.C.; Buboltz, J.R.;
Forte, C.C.; Bauer, A.C.; Manfro, R.C.; Souza, G.C.; Leitão, C.B. Effect of an intensive
nutrition intervention of a high protein and low glycemic-index diet on weight of kidney
transplant recipients: study protocol for a randomized clinical trial. Trials 2017, 18, 413,
doi:10.1186/s13063-017-2158-2.
- P3, assessment of the quality of life: please cite the reference
Response: We thank the reviewer for noticing this. Accordingly, we included the
following reference:
Hays, R.D.; Morales, L.S. The RAND-36 measure of health-related quality of life. Ann. Med.
2001, 33, doi:10.3109/07853890109002089.
- Finally, I did not see the supplementary materials. Would you please check it?
Response: We would like to provide our sincere apologies for the fact that the supplementary materials were not visible at the time of the review of this manuscript. It is not clear to us how this could happen. To ensure these materials are accessible for the evaluation of the revised version of the manuscript, we enclosed the supplementary tables in a separate file together with the revised version of the manuscript. The tracked changes in these supplementary tables are based on comment #3 of reviewer 2.
Reviewer 2 Report
This study uses data from “TransplantLines cohort and biobank study” from June 2015 to October 2019 to explore associations between protein intake and fatigue & quality of life (QoL) among “stable” kidney transplant recipients (KTR). The following points will help strengthen the manuscript.
1.Lines 77-83: Could authors discuss how a study that uses data from June 2015 to October 2019 could be called “cross-sectional”?
2. Lines 77-87: Could authors provide information about the study setting? How representative are those 730 participants of all “stable” KTR in Netherlands?
3. Lines 83-85: Could authors provide further information about the study sample: (a) how were 730 participants distributed across those 4+ study years, (b) are all of those 730 participants unique individuals, (c) as a related issue, is there a possibility that some of those 730 participants are not unique individuals because they were re-transplanted over those 4+ years, and (d) are all of those 730 participants kidney transplant recipients only or are some of those have received combined transplantation?
4. Lines 89-95: Could authors include a description of the time point at which protein intake assessment used in this study was conducted given that they report the data is coming from “TransplantLines cohort and biobank study” (indicating if protein intake assessments were undertaken at several time points for the cohort study and the specific time point used for the current study)?
5. Lines 97-106: Again, please be specific about the time point at which fatigue assessment used in this study was conducted and link this time point to the one used for protein intake assessment.
6. Lines 108-116: Again, please be specific about the time point at which QoL assessment used in this study was conducted and link this time point to the ones used for protein intake & fatigue assessments.
7. Lines 118-134: Similarly, please again be specific about the time points at which covariate assessments used in this study were conducted and link these time points to the ones used for protein intake, fatigue & QoL assessments.
8. Lines 136-160: As before, could authors discuss how data that spans 4+ years could be analyzed as if it was cross-sectional?
9. Line 247: “Stable” outpatient KTR is included in the title and mentioned in the Abstract (line 28) & Introduction (line 68) but is not mentioned again until the Discussion section. Could authors please define what “stable” means and how they identified “stable” KTR for their study?
10. Lines 313-314: Thanks for indicating the size of the study sample as being a strength. However, as indicated above please ensure that the statistical analyses are in line with the characteristics of the study sample that spans 4+ years.
Author Response
Dear Editor,
We thank the reviewers for the valuable comments and the opportunity to submit a revised version of our manuscript to be considered for publication. We have changed the manuscript according to the suggestions of all reviewers, to strengthen our manuscript. A detailed, itemized response to all comments can be found in this document. Please find enclosed our revised manuscript with ID nutrients-874426, entitled “Protein Intake, Fatigue and Quality of Life in Stable Outpatient Kidney Transplant Recipients.”
Sincerely,
On behalf of all the authors,
Karin Boslooper-Meulenbelt
This study uses data from “TransplantLines cohort and biobank study” from June 2015 to October 2019 to explore associations between protein intake and fatigue & quality of life (QoL) among “stable” kidney transplant recipients (KTR). The following points will help strengthen the manuscript.
1.Lines 77-83: Could authors discuss how a study that uses data from June 2015 to October 2019 could be called “cross-sectional”?
Response: We thank the reviewer for this comment to clarify the cross-sectional design of the study. In this study, all measurements took place during a single study visit of each of the participants, so the outcomes (fatigue and quality of life) and exposure (protein intake) were assessed at the same time point for each participant. The participants were sequentially included in the study from June 2015 until October 2019, to make data of a single measurement available in all participants.
To accommodate the comment of the reviewer, we added mentioning of this information to the revised version of the manuscript. The added sentences reads as follows: “All measurements took place during a single study visit of each of the participants, so the outcomes (fatigue and QoL) and exposure (protein intake) were assessed at the same time point for each participant.” (Chapter Materials and Methods, line 80 – 83 in the manuscript with tracked changes)
Lines 77-87: Could authors provide information about the study setting? How representative are those 730 participants of all “stable” KTR in Netherlands?
Response: Regarding the study setting, we included KTR who had a scheduled study visit at 1 year after transplantation or later after kidney transplantation. For 219 KTR, the study visit was at one year after transplantation and for 513 KTR it was at more than 1 year after transplantation.
To accommodate the comment of the reviewer, we added mentioning of this information to the revised version of the manuscript:
‘Of 219 KTR we used the data from the 1-year study visit and of 511 KTR we used the data of the study visit at more than 1 year after transplantation.’ (Chapter Materials and Methods, paragraph 2.1, line 101-102 in the manuscript with tracked changes)
Regarding the representativeness of the 730 KTR in the study compared to all KTR in the Netherlands, we are unfortunately not able to compare the baseline characteristics of our KTR population with other kidney transplant centers, as the TransplantLines Cohort and Biobank study is a single center study of the UMCG. However, all KTR from the North-East region of the Netherlands have an annual outpatient visit in the UMCG kidney transplant center, so the UMCG covers a large geographical area within the Netherlands. To examine the representativeness of the 730 KTR for the total KTR population of the UMCG, we compared the baseline characteristics (sex, age, eGFR and BMI) of the 730 included participating KTR with all non-participating KTR who visited the outpatient clinic between June 2015 and October 2019. These non-participants were either not yet invited for study participation or declined study participation.
We found no significant differences between participants and non-participants with respect to sex (58% male in participants versus 57% male in non-participants, P = 0.90), age (56 ± 13 years in participants versus 56 ± 14 years in non-participants, P = 0.87), estimated glomerular filtration rate (eGFR) (51.2 ± 17.9 mL/min/1.73m2 in participants versus 50.5 ± 19.5 mL/min/1.73m2 , P = 0.42) and Body Mass Index (BMI) (27.2 ± 4.7 kg/m2 in participants versus 26.9 ± 8.2 kg/m2 in non-participants, P = 0.23).
So, based on these findings the study sample is representative for KTR from the North-East region of the Netherlands. We acknowledge that use of data from a single center study is a limitation of the study. To accommodate the comment of the reviewer, we added mentioning of this information to the revised version of the manuscript:
‘Of a total of 2049 outpatient KTR visiting the outpatient clinic at least once yearly, 1034 KTR had been invited for study participation at the time of closure of the database for this study, of which 812 (78.5%) had signed informed consent and finalized the study visit. We excluded all participants with missing data on fatigue or protein intake (n=88), resulting in 730 participants eligible for analyses. No significant differences were found differences between the included participants and non-participants with respect to sex (58% male in participants versus 57% male in non-participants, P = 0.90), age (56 ± 13 years in participants versus 56 ± 14 years in non-participants, P = 0.87), estimated glomerular filtration rate (eGFR) (51.2 ± 17.9 mL/min/1.73m2 in participants versus 50.5 ± 19.5 mL/min/1.73m2 , P = 0.42) and Body Mass Index (BMI) (27.2 ± 4.7 kg/m2 in participants versus 26.9 ± 8.2 kg/m2 in non-participants, P = 0.23).’ (Chapter Materials and Methods, line 90-99 in the manuscript with tracked changes)
‘For testing differences in sex, age, eGFR and BMI of the included participants of this study versus the non-participants (paragraph 2.1), the independent t-test was used for normally distributed data and the chi-square test for categorical data.’ (Chapter Materials and Methods, line 163-165 in the manuscript with tracked changes)
In addition we added the following sentence to describe the limitation of a single center study: ‘Finally, as this is a single center study, we were not able to ascertain whether the KTR of our study population are representative of all KTR of the Netherlands. Similarly, we cannot ascertain whether our results can be extrapolated to KTR in general.’ (Chapter Discussion, line 366-369 in the manuscript with tracked changes)
- Lines 83-85: Could authors provide further information about the study sample: (a) how were 730 participants distributed across those 4+ study years, (b) are all of those 730 participants unique individuals, (c) as a related issue, is there a possibility that some of those 730 participants are not unique individuals because they were re-transplanted over those 4+ years, and (d) are all of those 730 participants kidney transplant recipients only or are some of those have received combined transplantation?
- Response: We thank the reviewer for this comment that allows for providing more
information of the study sample. The distribution of the study visits of the 730 KTR was as follows: 2015: 27, 2016: 180, 2017: 200, 2018: 230 and 2019: 93.
Accordingly, we added the following information to the revised version of the manuscript: ‘With regard to the distribution of the study visits across the study period, 27 study visits were performed in 2015, 180 in 2016, 200 in 2017, 230 in 2018 and 93 in 2019.’ (Chapter Materials and Methods, paragraph 2.1, line 103-104 in the manuscript with tracked changes)
b/c) all participants are unique individuals, which was verified by checking their unique research identification code. None of the KTR was included twice because of re-transplantation. Accordingly, the following sentence was added to the revised version of the manuscript: ‘All participants are unique individuals, which was verified by checking their unique research identification code. None of the KTR was included twice because of re-transplantation .’ (Chapter Materials and Methods, line 83-84 in the manuscript with tracked changes)
- d) Of all 730 participants, 20 received a combined organ transplantation, of whom 17 received a combined kidney-pancreas transplantation and 3 a combined kidney-liver transplantation. It appeared that KTR with a combined transplantation more often experienced severe fatigue (p=0.02). Accordingly, we added this significant determinant of severe fatigue to all multivariable models in which transplant-specific variables are included (models 3-6). Although this did not materially affect the independent association of protein intake with fatigue, the presence of a combined organ transplantation remained also significantly associated with the presence of severe fatigue in the final model (model 6), while the significant association of diabetes with fatigue was lost after adding this variable. We regarded this, therefore, to be a relevant finding to be mentioned in the Results section of the revised version of the manuscript.
Accordingly, the variable ‘history of combined organ transplantation’ was also added to the linear regression models examining the association of protein intake with quality of life. This did not materially affect our findings, except for the association of protein intake (in g/kg/d) with the mental composite summary score (MCS) of quality of life in the sensitivity analyses, where significance was lost in the multivariate analyses (model 5 and 6 of Supplementary Table S3).
To accommodate the comment of the reviewer, we incorporated the variable ‘history of combined organ transplantation’ in Supplementary Table S1, where we show the final cumulative model of the multinomial logistic regression of protein intake with fatigue in KTR. We also adapted the Odds Ratios, 95% confidence intervals and P-values of other variables according to the revised model if needed. Because we added the variable ‘history of a combined organ transplantation’ to multinomial regression models 3-6 of Table 2 and Supplementary Table S2, we corrected the Odds Ratios, 95% confidence intervals and P-values in these Tables if needed. And because we also added the variable “history of a combined organ transplantation” to the linear regression models 3-6 of Table 3 and Supplementary Table S3, we corrected the beta’s, 95% confidence intervals and P-values in these Tables if needed. By correcting these data in the tables, we noticed a few typing errors in lines 259-260 and lines 275 – 276, which we also corrected.
Based on this comment the following sentences were added to the revised version of the manuscript:
‘A combined organ transplantation was performed in 20 KTR, of which 17 received a combined kidney-pancreas transplantation and 3 a combined kidney-liver transplantation.’ (Chapter Results, Line 195-197 in the manuscript with tracked changes).
‘In the final cumulative model (model 6), higher BMI, a history of dialysis, glomerulonephritis as primary kidney disease and a history of combined organ transplantation were also associated with severe fatigue (Supplementary table S1)’ (Chapter Results, line 243-246 in the manuscript with tracked changes).
In addition, the following sentence was adapted: ‘KTR with severe fatigue also less often had a pre-emptive transplantation (P=0.001), received a kidney from a living donor less often (P=0.008), more often had proteinuria (P=0.01) and more often had a history of combined organ transplantation (P=0.02).’ (Chapter Results, line 220-222 in the manuscript with tracked changes)
We also added ‘a history of combined organ transplantation’ to paragraph 2.6 of the revised version of the manuscript, which now reads as follow: ‘In multivariable multinomial logistic regression analyses, we cumulatively adjusted the association of protein intake with categories of fatigue, i.e. no to mild fatigue, moderate fatigue and severe fatigue for potential confounders, including, age, sex and BMI (model 1), eGFR, proteinuria, primary kidney disease (model 2), time after transplantation, pre-emptive transplantation, living kidney donor, mycophenolic acid, ciclosporin use and a history of combined organ transplantation (model 3), diabetes, systolic blood pressure, cholesterol, and statin use (model 4), hemoglobin-, ferritin-, vitamin B12-, albumin and C-reactive protein (CRP) concentrations (model 5) and smoking status, alcohol use and level of education (model 6).’ (Chapter Materials and Methods, line 178 - 186 in the manuscript with tracked changes)
We also included the data on a history of combined organ transplantation in Table 1 on page 6 of the revised version of the manuscript as well as in the abstract on page 1.
Furthermore, as the findings of the association of protein intake with MCS of quality of life was changed by including the variable ‘a history of combined organ transplantation’ to the models, we adapted the following sentence in the revised version of the manuscript: ‘For QoL, protein intake in g/kg/d was significantly associated with PCS (β 0.62 per 0.1 g/kg/d increment; 95%CI 0.32 – 0.91, P<0.001), but not with MCS (β 0.19 per 0.1 g/kg/d increment; 95%CI -0.07 – 0.44, P=0.16) in univariable analyses. These findings remained materially unchanged after adjustment for the several potential confounders (Supplementary Table S3).’ (Chapter Results, line 278 – 284 in the manuscript with tracked changes)
Finally, we incorporated these findings in the discussion section of the revised version of the manuscript: ‘Besides the association with protein intake, higher BMI, history of dialysis, glomerulonephritis as primary kidney disease, and a history of combined organ transplantation were also independently associated with severe fatigue in this cohort.’ (Chapter Discussion, line 294 – 297 in the manuscript with tracked changes)
Lines 89-95: Could authors include a description of the time point at which protein intake assessment used in this study was conducted given that they report the data is coming from “TransplantLines cohort and biobank study” (indicating if protein intake assessments were undertaken at several time points for the cohort study and the specific time point used for the current study)?
Response: We thank the reviewer for this comment and agree that the time point of the protein intake assessment should be described in more detail. For this study the data from blood and urine samples and the questionnaires was collected during a single study visit, either at 1 year after transplantation or later after transplantation. The median time point of all study visits was 4.1 [IQR 1.0 – 11.0] years after transplantation.
Based on this comment the we added the words “scheduled study” to the following sentence of the revised version of the manuscript: ‘All subjects were instructed to collect a 24-hour urine sample according to a strict protocol at the day before their scheduled study visit at the outpatient clinic.’(Chapter Materials and Methods, line 109-110 in the manuscript with tracked changes)
Accordingly, we also added the following sentence to the revised version of the manuscript (also see our response to comment #2 of this reviewer): ‘Of 219 KTR we used the data from the 1-year study visit and of 511 KTR we used the data of the study visit at more than 1 year after transplantation’ (Chapter Material and Methods, paragraph 2.1, line 101 – 102 in the manuscript with tracked changes)
Finally we adapted the following sentence in the revised version of our manuscript: ‘We included 730 KTR with a study visit at a median of 4.1 [IQR 1.0 – 11.0] years after transplantation.’ (Chapter Results, line 193-194 in the manuscript with tracked changes) and added decimals to the time since transplantation in Table 1 on page 6 of the manuscript with tracked changes.
5. Lines 97-106: Again, please be specific about the time point at which fatigue assessment used in this study was conducted and link this time point to the one used for protein intake assessment.
Response: This assessment took place at the same study visit when the blood and urine samples were collected. Accordingly, the following sentence was added to the revised version of the manuscript: ‘This study visit took place at the same time of the blood and urine samples collection.’ (Chapter Materials and Methods, line 119-120 in the manuscript with tracked changes)
Lines 108-116: Again, please be specific about the time point at which QoL assessment used in this study was conducted and link this time point to the ones used for protein intake & fatigue assessments.
Response: This assessment took place concomitant with the fatigue assessment and blood and urine sample collection at the same study visit. Accordingly, the following sentence was adapted in the manuscript: ‘The RAND-36 Health Survey (RAND-36) was used to evaluate QoL and was completed by the participants concomitant with the CIS-questionnaire during the same study visit.’ (Chapter Materials and Methods, line 130 -131 in the manuscript with tracked changes).
Lines 118-134: Similarly, please again be specific about the time points at which covariate assessments used in this study were conducted and link these time points to the ones used for protein intake, fatigue & QoL assessments.
Response: we agree with the reviewer we should describe the time points of covariate assessment in more detail. All covariate assessments took place during the same study visit as the assessment of protein intake, fatigue and QoL. An exception are the baseline data on primary kidney disease, dialysis treatment and donor organ specifics. These data were retrieved separately from the UMCG Renal Transplant Database.
Accordingly, the following sentence were adapted in the revised version of the manuscript: ‘Data of the covariates were collected during the same study visit, except for the information on primary kidney disease, dialysis treatment and donor organ specifics. These data were retrieved separately from the UMCG Renal Transplant Database.’ (Chapter Materials and Methods, line 141 – 143 in the manuscript with tracked changes)
Lines 136-160: As before, could authors discuss how data that spans 4+ years could be analyzed as if it was cross-sectional?
Response: We thank the reviewer for this comment to clarify the cross-sectional design of the study. In this study, all measurements took place during a single study visit of each participant, so the outcomes (in this study fatigue and quality of life) and exposure (in this study protein intake) were assessed at the same time point for each participant. The participants were sequentially included in the study from June 2015 until October 2019, and the study visits of all included participants took place within this time frame, to make data of on a single time point available in all participants. We kindly refer to our answer at comment #1 of this reviewer for the modifications in the revised version of the manuscript that clarifies the cross-sectional design of our study.
Line 247: “Stable” outpatient KTR is included in the title and mentioned in the Abstract (line 28) & Introduction (line 68) but is not mentioned again until the Discussion section. Could authors please define what “stable” means and how they identified “stable” KTR for their study?
Response: we thank the reviewer for this comment. For this study we defined ‘stable’ KTR as KTR ≥ 1 year after transplantation with a functioning graft and without known or apparent systemic illnesses, e.g. malignancies or opportunistic infection. This comment made us realize that this was not clearly described in the Materials and Methods.
Accordingly, the following sentences were adapted in the revised version of our manuscript: ‘For this study we included adult (≥18 years old) stable outpatient KTR, which was defined as having a functioning graft ≥ 1 year after transplantation without known or apparent systemic illnesses (i.e., malignancies, opportunistic infections). We included KTR at a scheduled study visit between June 2015 and October 2019.’ (Chapter Materials and Methods, line 85 – 89 in the manuscript with tracked changes)
Lines 313-314: Thanks for indicating the size of the study sample as being a strength. However, as indicated above please ensure that the statistical analyses are in line with the characteristics of the study sample that spans 4+ years.
Reply: We thank the reviewer for this comment to ensure that we used the correct statistical methods for the data analysis. As both the outcome variables (fatigue and quality of life) and exposure variables (protein intake), and covariates, were collected at the same time point, the data comprise the design of a cross-sectional study and the data were analyzed accordingly. To further ensure that we used the correct statistical tests, we consulted a senior statistician, who confirmed this.
Round 2
Reviewer 2 Report
Thanks to authors for detailed responses to review points; this is an improved manuscript.
All of the following 3 points below (old numbering is kept for convenience) refer to the same characteristic of the study data. The fact that the measurements for each participant were taken at a single point in time indicate how well designed the study is but, this reviewer still thinks that, this does not preclude the fact that the study data itself spans 4+ years. One suggestion is add a section to the Discussion section explaining what authors think about internal (such as if anything has changed in terms of transplantation or, in general, in terms of health care provision in their center over those years) and external (such as secular trends) factors, if any, that could have an impact on their results (given that the study data that spans 4+ years were analyzed as if it were cross-sectional).
1) Old Lines 77-83: Could authors discuss how a study that uses data from June 2015 to October 2019 could be called “cross-sectional”?
8) Old Lines 136-160: As before, could authors discuss how data that spans 4+ years could be analyzed as if it was cross-sectional?
10) Old Lines 313-314: Thanks for indicating the size of the study sample as being a strength. However, as indicated above please ensure that the statistical analyses are in line with the characteristics of the study sample that spans 4+ years.
Author Response
Dear Editor,
We thank the second reviewer for the additional comment and the opportunity to submit a revised version of our manuscript to be considered for publication. We have changed the manuscript according to the additional suggestion of the reviewer, to further strengthen our manuscript. The response to the comment of this reviewer can be found in this document. Please find enclosed our revised manuscript with ID nutrients-874426, entitled “Protein Intake, Fatigue and Quality of Life in Stable Outpatient Kidney Transplant Recipients.”
Sincerely,
On behalf of all the authors,
Karin Boslooper-Meulenbelt
Comment reviewer
All of the following 3 points below (old numbering is kept for convenience) refer to the same characteristic of the study data. The fact that the measurements for each participant were taken at a single point in time indicate how well designed the study is but, this reviewer still thinks that, this does not preclude the fact that the study data itself spans 4+ years. One suggestion is add a section to the Discussion section explaining what authors think about internal (such as if anything has changed in terms of transplantation or, in general, in terms of health care provision in their center over those years) and external (such as secular trends) factors, if any, that could have an impact on their results (given that the study data that spans 4+ years were analyzed as if it were cross-sectional).
1) Old Lines 77-83: Could authors discuss how a study that uses data from June 2015 to October 2019 could be called “cross-sectional”?
8) Old Lines 136-160: As before, could authors discuss how data that spans 4+ years could be analyzed as if it was cross-sectional?
10) Old Lines 313-314: Thanks for indicating the size of the study sample as being a strength. However, as indicated above please ensure that the statistical analyses are in line with the characteristics of the study sample that spans 4+ years.
Response: we thank the reviewer for the additional comment with the suggestion to add a mentioning of the fact that internal and external factors may have impacted our results given the 4 year time span of the data collection to the Discussion section of the revised version of the manuscript. To accommodate this comment of the reviewer we added this information to the revised version of the manuscript: : ‘Second, the data collection for this cross-sectional analysis took place in a time span of > 4 years. It cannot be excluded that this impacted the results by changes in transplant care or other, external factors within this time frame. It should, however, be noted that in this time frame there were neither changes in the immunosuppressive regimen nor in supportive care, including e.g. the antiviral and antimicrobial prophylaxis. In addition, other aspects of kidney transplant care and the health care system also remained materially unchanged.’ (Chapter Discussion, line 364 - 369 of the manuscript with tracked changes)